# Milk Production and Enteric Methane Emissions in Dairy Cows Grazing Annual Ryegrass Alone or Intercropped with Forage Legumes

**DOI:** 10.3390/ani15162329

**Published:** 2025-08-08

**Authors:** Larissa Godeski Moreira, Tiago Celso Baldissera, Chrystian Jassanã Cazarotto, Maria Isabel Martini, Renata da Rosa Dornelles, Henrique M. N. Ribeiro-Filho

**Affiliations:** 1Centro de Ciências Agroveterinárias, Departamento de Produção Animal e Alimentos, Universidade do Estado de Santa Catarina, Lages 88520-000, SC, Brazil; larissa.moreira@edu.udesc.br (L.G.M.); cj.cazarotto@edu.udesc.br (C.J.C.); maria.martini@edu.udesc.br (M.I.M.); renata.dornelles@edu.udesc.br (R.d.R.D.); 2Empresa de Pesquisa Agropecuária e Extensão Rural de Santa Catarina, Lages 88502-970, SC, Brazil; tiagobaldissera@epagri.sc.gov.br

**Keywords:** methane, dairy cows, grazing, *Lolium multiflorum* Lam., *Trifolium pratense* L., *Vicia sativa* L.

## Abstract

This study explored whether adding forage legumes—common vetch and red clover—to annual ryegrass pastures, while reducing nitrogen fertilizer use, could improve milk production and lower methane emissions from grazing dairy cows. Twelve cows grazed either pure ryegrass or a mixture of ryegrass and legumes, and researchers measured their milk output, feeding behavior, and methane emissions. The legumes contributed only about 9% of the total pasture, and cows grazing the mixed pastures had access to less available forage. There were no significant differences in milk production, daily methane emissions, or methane produced per unit of milk between the two groups. These results suggest that simply reducing nitrogen fertilizer and adding a small amount of legumes may negatively affect sward structure and may be insufficient to enhance productivity or mitigate environmental impacts. To make pasture-based dairy farming more sustainable, more effective strategies are needed to increase both the proportion of legumes and the total forage yield in mixed pastures.

## 1. Introduction

Pasture-based dairy systems are increasingly acknowledged as a promising paradigm for sustainable milk production, particularly when supported by the availability of high-quality forages. These systems not only optimize the conversion of natural resources into nutrient-rich dairy products, but also align environmental responsibility with economic efficiency by reducing the reliance on imported feeds, minimizing greenhouse gas emissions, and enhancing nutrient-use efficiency [1,2]. In such systems, maximizing forage utilization and quality is essential to support productive and environmentally efficient animals. Annual ryegrass (*Lolium multiflorum* Lam.) is widely used due to its high nutritive value [3] and productivity [4] in temperate and subtropical regions; however, it still requires significant nitrogen (N) inputs and may present limitations in crude protein (CP) concentration and digestibility, particularly in later growth stages [5,6].

The inclusion of forage legumes such as clovers and vetches in grass swards has been proposed as a strategy to improve herbage nutritive value, increase milk production, and reduce environmental burdens [7]. Legumes contribute to biological nitrogen fixation, which can reduce fertilizer use, and often exhibit higher protein content and lower fiber fractions [8]. Additionally, certain legumes may reduce enteric methane (CH_4_) production by altering rumen fermentation kinetics or through the action of secondary metabolites [9].

Despite these potential advantages, successful incorporation of legumes into intensively managed pastures remains challenging. Factors such as interspecific competition [10], seasonal establishment constraints [11], and grazing pressure often limit legume contribution to sward biomass [12]. Consequently, the expected improvements in animal performance and emissions mitigation may not be realized under practical grazing scenarios.

This study aimed to investigate the effects of reducing nitrogen fertilization and intercropping annual ryegrass with common vetch (*Vicia sativa* L.) and red clover (*Trifolium pratense* L.) on productive and environmental parameters in pasture-based systems for early-lactation dairy cows. We hypothesized that the inclusion of legumes would enhance herbage quality and reduce CH_4_ intensity (g/kg ECM) without negatively affecting milk yield.

## 2. Materials and Methods

All procedures involving animals were approved by the Animal Ethics Committee of the State University of Santa Catarina (protocol number 6051180521).

### 2.1. Experimental Area and Treatments

The experiment was conducted in Lages, Santa Catarina, Brazil (27°47′ S; 50°18′ W; 920 m altitude), from September to November 2023. A 4.2 ha experimental area was divided into two subplots (monoculture ryegrass: RG; ryegrass intercropped with legumes: RG + Leg), each subdivided into paddocks for adaptation and data collection.

Treatments involved annual ryegrass (*Lolium multiflorum* Lam.) either as a monoculture or intercropped with common vetch (*Vicia sativa* L.) and red clover (*Trifolium pratense* L.). Ryegrass was sown at 40 kg/ha and legumes at 6 kg/ha. Nitrogen application was 150 kg/ha for RG and 75 kg/ha for RG + Leg. Rotational grazing was employed (four days of occupation), targeting 40–50% defoliation. Twelve early-lactation Holstein × Jersey cows (66.5 ± 24.6 days in milk) were assigned to treatments using a crossover design (two 12-day periods: eight for adaptation, four for data collection). Cows were supplemented daily with 4 kg of ground sorghum (849 g DM/kg, 917 g OM/kg DM, 110 g CP/kg DM, 145 g NDF/kg DM, and 52 g ADF/kg DM) and 160 g of a mineral mix.

### 2.2. Pasture Measurements

Pasture height was measured before and after grazing in each period. In the first period, 100 height measurements per paddock were taken using a sward stick, and 200 compressed height readings were taken with a rising plate meter (Farmworks^®^, model F200, Palmerston North, New Zealand). Pre-grazing herbage mass was estimated based on a calibration curve from the plate meter, developed using five sample points where compressed height was recorded and aboveground biomass was harvested from 0.1 m^2^ quadrats. After cutting, the samples were weighed and dried in a forced-air oven at 60 °C for 72 h. Herbage mass (kg DM/ha) for the ryegrass-only and ryegrass-legume treatments was estimated using the following equations:

Equation (1) (monoculture ryegrass):y = 142 − 340 × pre-grazing compressed height (cm) (R^2^ = 0.83)(1)

Equation (2) (intercropped):y = 215 − 1008 × pre-grazing compressed height (cm) (R^2^ = 0.89)(2)

In the second period, due to a technical failure of the rising plate meter, pre-grazing forage mass was estimated by destructive sampling, using 20 quadrats (0.24 m^2^ each; 0.8 × 0.3 m) per paddock. Samples were dried under the same conditions previously described for dry matter determination.

The chemical and botanical composition was assessed using samples collected the day before grazing in each period. At least 20 samples per paddock were harvested at ground level with hand shears, within a ~20 cm diameter, forming a composite sample. The samples were frozen at −20 °C and, after thawing, split into two subsamples: one for botanical separation and the other cut at the average post-grazing tiller height, with the upper part used for chemical analysis of consumed forage. Post-grazing tiller height was measured on 160 tillers per paddock, using a graduated ruler.

### 2.3. Animal Measurements

Milk yield and composition were evaluated during the last four days of each period. Milk production was recorded daily during both milkings (7 am and 3 pm), using an electronic meter (Waikato Milking Systems, Hamilton, New Zealand). Milk samples from both milkings were collected daily on the same days that milk production was recorded, for subsequent chemical composition analysis.

Methane emissions were measured using two GreenFeed systems (C-Lock Inc., Rapid City, SD, USA). Each cow accessed the system on two separate days—specifically on days 9 and 11 of each experimental period—ensuring gas sampling every 3 h over a 24 h cycle. On the second day of sampling, the feeding schedule was shifted 3 h earlier to capture different diurnal patterns. To ensure data reliability, each cow remained at the GreenFeed unit for a minimum of 4 min per visit, with successful measurements obtained at least once at the following time points: 6:30 h, 9:30 h, 12:30 h, 15:30 h, 18:30 h, and 21:30 h.

Grazing behavior was visually and systematically recorded every five minutes on days 10 and 12 of each period, following the methodology of Penning [13]. Observations were conducted throughout the daytime grazing cycle, categorizing activities as grazing, ruminating, or idling. These data allowed for estimation of the time and daily proportion spent on each activity.

Herbage digestibility was estimated using the chemical composition of feces and consumed forage. Fecal samples were collected after each milking for four consecutive days. After drying (72 h at 60 °C), samples were ground (1 mm) for analysis. Organic matter digestibility (OMD, g/g OM) was estimated using the equation by Ribeiro-Filho et al. [14]:OMD = 1.035 − 24.78/CPf − 0.00027 × ADFf − 0.0571 × CPh/CPf(3)
where CPf = fecal crude protein (g/kg OM), ADFf = fecal acid detergent fiber (g/kg OM), and CPh = forage crude protein (g/kg OM). Methane (CH_4_) emissions were measured on two of the last four days of each period, on days different from those used for behavior observations.

### 2.4. Chemical Analyses

Dry matter (DM) content was determined by drying at 105 °C for 24 h. Ash content was obtained via combustion in a muffle furnace at 550 °C for 4 h, and organic matter (OM) was calculated by mass difference. Total nitrogen (N) content was determined using the Dumas combustion method 968.06 [15] with a Leco FP 528 instrument (Leco Corporation, Saint Joseph, MI, USA). Crude protein (CP) was calculated by multiplying N content by 6.25. Neutral detergent fiber (NDF) was determined according to Mertens [16], with modifications: samples were weighed in filter bags and treated with neutral detergent in an ANKOM A220 system (ANKOM Technology, Macedon, NY, USA). This analysis included thermostable α-amylase and ash correction but excluded sodium sulfite. Acid detergent fiber (ADF) was analyzed according to AOAC Method 973.18 [15].

### 2.5. Statistical Analysis

All statistical analyses were conducted using the MIXED procedure of SAS 9.4 (SAS Institute Inc., Cary, NC, USA). A linear mixed model for repeated measures was used to account for the correlated nature of data collected over time on the same experimental units (i.e., cows). The model included the covariate herbage allowance, the fixed effect of treatment, and random effects of period and cow within period. Day was treated as a repeated measure, and its interaction with treatment was also included as a fixed effect. The statistical model used wasY_ijkl_ = μ + βHA_ijkl_ + treatment_i_ + period_j_ + cow_k_ (p_i_) + day_l_ + (treatment × day)_il_ + e_ijkl_
where Y_ijk_, μ, βHA_ijkl_ + treatment_i_ + period_j_ + cow_k_ (p_i_) + day_l_ + (treatment × day)_il_, and e_ijkl_ represent the analyzed variable, the overall mean, the linear effect of herbage allowance (included as continuous covariate), the fixed effect of treatment, the random effect of period, the random effect of cow nested within period, the fixed effect of day, the treatment × day interaction, and the residual error term.

Repeated measures were specified across day within cow, and alternative covariance structures (compound symmetry, autoregressive AR(1), and unstructured) were tested. The structure with the lowest Akaike Information Criterion (AIC) and Bayesian Information Criterion (BIC) was selected for the final model. Least squares means (LSMEANS) were obtained and compared using the Tukey–Kramer adjustment for multiple comparisons. Statistical significance was declared at *p* < 0.05.

## 3. Results

Pre-grazing herbage mass averaged around 1800 kg DM/ha in RG paddocks and approximately 1500 kg DM/ha in RG + Leg paddocks, while herbage allowance exceeded 50 kg DM/cow/day in RG and was close to 46 kg DM/cow/day in RG + Leg paddocks (Table 1). Legumes accounted for ~9% of the DM in RG + Leg treatment. Crude protein content was slightly higher in RG + Leg (178 vs. 172 g/kg DM), while NDF and ADF were also marginally greater.

Daily CH_4_ emissions and emission intensity did not differ between treatments, averaging 329 g/day and 11.4 g/kg ECM (Table 2). Milk production, ECM, milk fat, protein concentration, and milk urea nitrogen (MUN) were similar between treatments. Moreover, none of these variables showed significant differences across the evaluation days or in the treatment × day interaction, indicating that responses remained stable over time in both treatments.

Grazing time averaged ~390 min/day with no major differences between treatments (Table 3). Cows grazed most after afternoon milking (16:00 h–20:00 h, 34%), followed by post-morning milking (8:00 h–12:00 h, 28%). Additionally, grazing behavior remained consistent throughout the measurement period, with no significant effects of day or treatment × day interaction, suggesting a stable daily pattern across both treatments.

## 4. Discussion

Contrary to the hypothesis, intercropping with red clover and vetch did not improve milk yield or methane output. Instead, milk production, milk composition, and methane emissions remained unchanged, likely due to the low legume intake observed during grazing, which may have limited the functional impact of legume inclusion.

The proportion of legumes in the intercropped pastures accounted for approximately 9% of the total dry matter, indicating limited competitiveness with ryegrass and likely contributing to the reduced overall dry matter yield. Other studies reporting benefits from legume inclusion have typically achieved higher proportions (≥20%) in the sward, often through the use of more aggressive species or modified management practices [18,19].

The similar milk production in cows grazing monoculture ryegrass pastures and cows on intercropped pastures likely reflects a compensation due to higher herbage mass and herbage allowance in the monoculture treatment. Lower herbage mass in mixed swards, such as those containing white clover and perennial ryegrass, can lead to reduced herbage intake and milk yield. For instance, cows grazing on mixed swards with lower pre-grazing herbage mass had lower herbage intake and milk yield compared to those grazing on perennial ryegrass monocultures [14]. This suggests that the structure and availability of herbage in mixed swards can significantly impact intake and subsequent milk production. In the same way, it is well known that lower herbage allowance can decrease individual milk production, with reductions of approximately 9.5% observed when daily herbage allowance decreases from 30 to 20 kg of dry matter per cow [20,21].

The lack of difference in milk fat, ECM, and urea nitrogen concentrations suggests that the cows’ metabolic responses were similar, but that reduced intake or forage quality limited output in the mixed treatment. The inclusion of legumes in grass swards generally improves the nutritional quality of the pasture, which can enhance milk production [22]. However, if the herbage mass is insufficient, the potential benefits of the mixed sward are not fully achieved. Cows grazing on mixed swards with adequate herbage mass and herbage allowance showed improved milk yield compared to those on monocultures, but this advantage diminishes with lower herbage mass [23,24].

The absence of a significant effect of legume inclusion on daily CH_4_ emissions and emission intensity may be partly explained by the low proportion of legumes in the sward. The methane-mitigating potential of legumes is generally associated with higher legume proportions—typically exceeding 20% of the sward dry matter [17,18]—as well as the presence of tannins or other bioactive compounds [25]. In the present study, the limited legume biomass likely restricted any potential advantages of the mixed swards. Although legumes can reduce methane production due to their physical and chemical properties, such effects are largely attributed to increased digesta passage rates and specific chemical interactions, rather than fundamental shifts in ruminal fermentation pathways [25,26,27].

Daily methane emissions from dairy cows grazing on temperate pastures, including grass–clover pastures, typically range from 151 to 497 g/day, with an average of 311 g/day [28]. This aligns closely with the average emission observed in the current study, which is 329 g/day. However, when considering methane intensity, dairy cows on temperate pastures exhibit values ranging from 19.9 to 26 g/kg ECM [29]. This intensity is substantially higher than the 11.4 g/kg ECM documented in our study. This apparent discrepancy can be attributed to the early lactation stage of the cows in our experiment, which had an average ECM yield of 30.5 kg/day. In contrast, the study by Moate et al. [29] encompassed cows throughout their entire lactation period, with milk production varying between 2889 to 7500 kg. Therefore, the differences in methane intensity can be largely explained by the lactation stage and the associated milk production levels of the cows evaluated.

Grazing behavior patterns were broadly similar between treatments, indicating that short-term behavioral responses to changes in forage composition were minimal under the conditions tested. It is well known that factors such as herbage allowance, grazing management practices, and specific structural characteristics of swards can play a role, but their effects are often context-dependent and may not lead to substantial changes in overall grazing behavior or animal performance [30,31]. Additionally, under high herbage allowance conditions, cows are able to graze selectively, which may reduce the influence of sward structure on grazing behavior [32]. In the present study, the daily herbage allowance exceeded 45 kg DM/cow, a level typically associated with lenient grazing management [33]. However, under such lenient grazing conditions, it is unclear whether cows effectively consumed significant amounts of the legume component, as post-grazing botanical composition was not assessed. This represents a limitation of the study, and further research under more typical or controlled grazing scenarios is needed to confirm these findings.

The limited benefits observed with legume inclusion are likely attributable to the low proportion of legumes in the sward. To achieve measurable gains in animal performance or environmental indicators, strategies that enhance legume establishment and persistence are essential. These may include species selection [34], adjusted sowing ratios [35], and grazing regimes [36] that favor legume competitiveness.

## 5. Conclusions

This study highlights that the potential of forage legumes to improve productivity and mitigate methane emissions in pasture-based dairy systems is closely linked to their proportion in the sward. Achieving effective legume establishment and persistence remains a critical challenge that warrants further research under rotational grazing conditions in subtropical environments.

## Figures and Tables

**Table 1 animals-15-02329-t001:** Herbage characteristics of annual ryegrass (*Lolium multiflorum* Lam.) alone or annual ryegrass + legumes.

	RG	RG + Leg
Pre-grazing herbage mass (kg DM/ha)	1773	1549
Herbage allowance (kg DM/cow)	51.7	45.8
Pre-grazing herbage height (cm)	32.4	28.2
Post-grazing herbage height (cm)	18.2	16.9
Defoliation severity ^1^	0.44	0.40
Chemical composition of the offered forage (g/kg DM)		
Dry matter (g/kg fresh)	144	160
Organic matter	897	901
Crude protein	172	178
Neutral detergent fiber	562	594
Acid detergent fiber	248	260
Botanical composition of the offered forage (g/kg DM)		
Ryegrass		
Leaves	434	398
Stems	357	342
Legumes	0.0	89.0
Other species	0.82	19.0
Dead material	205	151

^1^ Defoliation severity: (pre-grazing herbage height − post-grazing herbage height)/pre-grazing herbage height.

**Table 2 animals-15-02329-t002:** Methane emissions, milk yield, and milk composition of dairy cows grazing annual ryegrass (RG) or ryegrass + legumes (RG + Leg).

				*p*-Value		
	RG	RG + Leg	*rsd*	Treat	Day	Treat × Day
Methane						
g/day	335	323	48.9	0.679	0.128	0.722
g/kg ECM ^1^	11.2	10.6	1.63	0.577	0.202	0.816
Milk production (kg/day)	30.9	30.2	4.19	0.761	0.260	0.975
ECM production (kg/day)	30.3	30.6	3.90	0.865	0.174	0.969
Milk fat (g/kg)	38.9	41.1	3.66	0.317	0.226	0.780
Milk protein (g/kg)	33.2	32.7	3.10	0.783	0.392	0.769
Milk fat production (g/day)	1201	1237	176.9	0.726	0.114	0.942
Milk protein production (g/day)	1021	983	128.0	0.620	0.271	0.896
MUN ^2^ (mg/dL)	6.3	6.4	1.57	0.875	0.302	0.104
Live weight (kg)	535	574	68.9	0.354	0.909	0.901
Digestibility of OM (g/g OM)	0.809	0.802	0.0106	0.309	0.898	0.891

^1^ Energy-corrected milk production, calculated as follows: kg of milk production × [37.6 × fat (g/kg) + 20.9 × protein (g/kg) + 948]/3138 [17], ^2^ MUN: milk urea nitrogen. *rsd*: residual standard deviation.

**Table 3 animals-15-02329-t003:** Grazing time of dairy cows grazing annual ryegrass (RG) or ryegrass + legumes (RG + Leg).

				*p*-Value		
	RG	RG + Leg	*rsd*	Treat	Day	Treat × Day
Grazing time (min)						
5:00 h–8:00 h	63	51	14.7	0.164	0.113	0.800
8:00 h–12:00 h	110	107	13.9	0.776	0.895	0.232
12:00 h–16:00 h	93	87	15.9	0.516	0.101	0.162
16:00 h–20:00 h	131	136	8.3	0.324	0.199	0.545
Total	397	382	27.6	0.353	0.433	0.684

*rsd*: residual standard deviation.

## Data Availability

Data is available upon request.

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
