# Peer review of "Milk Production and Enteric Methane Emissions in Dairy Cows Grazing Annual Ryegrass Alone or Intercropped with Forage Legumes"

_animals, 2025, doi:10.3390/ani15162329_

Round 1
Reviewer 1 Report
Comments and Suggestions for Authors
This was an interesting study of lactating dairy cows on two different sward types. It is at very small scale and the authors do not overstate their results. A few comments are below. The legume consumption should be mentioned in the discussion.
L122 For clarity please specify: "Each animal accessed the system on two different days"
Table 2. Spelling of protein and day
L234. Under such lenient grazing it is not clear from the study whether the cows consumed much of the legume, such as from a post-grazing botanical composition. I suggest this is a limitation of the study which should be included here in the discussion. Further work with more complete typical grazing could confirm these results.
Author Response
Comments 1: This was an interesting study of lactating dairy cows on two different sward types. It is at very small scale and the authors do not overstate their results. A few comments are below. The legume consumption should be mentioned in the discussion.
Response 1: We sincerely thank you for your positive assessment and thoughtful feedback. In response to your suggestion, we included a comment on legume consumption in the Discussion section to highlight this limitation (Line 201 to 2043 in the revised manuscript)
Comments 2: L122 For clarity please specify: "Each animal accessed the system on two different days"
Response 2: The sentence was rephrased for clarity, as suggested (Line 121 to 123 in the revised manuscript).
Comments 3: Table 2 Spelling of protein and day
Response 3: The spelling errors in the table were corrected as indicated.
Comments 4: L234. Under such lenient grazing it is not clear from the study whether the cows consumed much of the legume, such as from a post-grazing botanical composition. I suggest this is a limitation of the study which should be included here in the discussion. Further work with more complete typical grazing could confirm these results.
Response 4: A sentence was added in the Discussion to acknowledge this limitation, as recommended. We agree that further studies under more typical grazing conditions are needed to validate these findings (Line 258 to 262 in the revised manuscript).
Reviewer 2 Report
Comments and Suggestions for Authors
An article on milk production and enteric methane emissions in dairy cows grazing annual ryegrass alone or intercropped with forage legumes was conducted with a clear hypothesis aligned with the methodology. However, the treatment that different pasture types received confounded the effects of forage availability for cows. The research should try to reduce the difference before testing the effect of pasture type. Lowering the available forage for cows exactly must affect milk production as well as methane emission; then, that result can't be identified from which factors. To improve this manuscript, the availability of forage for cows should be added to the statistical model as a fixed effect.
One more thing, Line 105 stated that a technical failure of the rising plate meter occurred in the second period. Does this impact the result of this study or not?
Author Response
Comments 1: An article on milk production and enteric methane emissions in dairy cows grazing annual ryegrass alone or intercropped with forage legumes was conducted with a clear hypothesis aligned with the methodology. However, the treatment that different pasture types received confounded the effects of forage availability for cows. The research should try to reduce the difference before testing the effect of pasture type. Lowering the available forage for cows exactly must affect milk production as well as methane emission; then, that result can't be identified from which factors. To improve this manuscript, the availability of forage for cows should be added to the statistical model as a fixed effect.
Response 1: Thank you for your insightful comment. In response, herbage allowance was included in the statistical model as a covariate. The analysis was repeated accordingly, and adjusted means, residual standard deviations, and p-values have been updated in the tables and throughout the manuscript.
Comments 2: One more thing, Line 105 stated that a technical failure of the rising plate meter occurred in the second period. Does this impact the result of this study or not?
Response 2: We agree that consistency in measurement methods is ideal. Although a technical issue prevented the use of the rising plate meter in the second period, we took great care to ensure the reliability of herbage mass estimates and to minimize any potential bias.
Reviewer 3 Report
Comments and Suggestions for Authors
Keywords: Review the keywords.
Abstract: Please review the abstract and replace or remove some sentences that are not relevant to this topic.
Results:Lines 160 - Do not use informal symbols.
Lines 159-162 - Be more direct in the results.
Lines 173-174 - Trend?
Recommended to use normal hours.
Lines 178-179 - It is not necessary to repeat what has been done.
Table 3 - Use standard hours (line 5 of the table)
Discussion: In this section, it would be interesting for the author's justification to always come first, explaining the results, and in the background, the citations supporting this justification.
Conclusion: Linhas 245-246 - In the conclusion, it is not appropriate to repeat results. Please review the conclusion and replace or remove some sentences that are not relevant. Consider summarizing the main findings in one or two sentences, highlighting only the most relevant results and their direct implications.
Author Response
Comments 1: Abstract: Please review the abstract and replace or remove some sentences that are not relevant to this topic.
Response 1: The abstract and keywords were revised to enhance clarity and focus. Sentences not directly relevant to the core findings were removed or rephrased.
Comments 2: Results: Lines 160 - Do not use informal symbols.
Response 2: The word “treatment” was added for clarity.
Comments 3: Lines 159-162 - Be more direct in the results.
Response 3: The sentence was rephrased to present the results more directly and concisely.
Comments 4: Lines 173-174 - Trend?
Response4: The sentence was removed as suggested.
Comments 5: Recommended to use normal hours.
Response 5: All time references were converted to standard 24-hour format.
Comments 6: Lines 178-179 - It is not necessary to repeat what has been done.
Response 6: The redundant sentence was removed.
Comments 7: Table 3 - Use standard hours (line 5 of the table)
Response 7: Standard (24-hour) time format was used in Table 3 as recommended.
Comments 8: Discussion: In this section, it would be interesting for the author's justification to always come first, explaining the results, and in the background, the citations supporting this justification.
Response 8: Several paragraphs in the Discussion section were revised to follow this structure. The interpretation of results was placed before citations to improve logical flow and reader understanding.
Comments 9: Conclusion: Linhas 245-246 - In the conclusion, it is not appropriate to repeat results. Please review the conclusion and replace or remove some sentences that are not relevant. Consider summarizing the main findings in one or two sentences, highlighting only the most relevant results and their direct implications.
Response 9: The conclusion section was rephrased to avoid repeating specific results and to clearly highlight the key implication of the study.
Reviewer 4 Report
Comments and Suggestions for Authors
While I find the research very interesting and the topic highly relevant within the context of pasture-based systems, there are several aspects that are insuficiently elaborated, or not clearly presented on the current version of the manuscript. I encourage the authors to address this points in order to strengthen the scientific rigour and clarity of this work.
Please find attached the revised version of the manuscript with the suggestions and questions inserted as comments.

The manuscript would benefit from a revision of the English language to improve clarity and grammar. While the overall meaning is generally fine, there are some instances where imprecise wording may hinder comprehension. Some suggestions for improvement have been made directly in the reviewed version of the manuscript (inserted as a comment).
Author Response
Comments: While I find the research very interesting and the topic highly relevant within the context of pasture-based systems, there are several aspects that are insuficiently elaborated, or not clearly presented on the current version of the manuscript. I encourage the authors to address this points in order to strengthen the scientific rigour and clarity of this work.
Please find attached the revised version of the manuscript with the suggestions and questions inserted as comments.
Response: We sincerely appreciate the reviewer’s thoughtful and constructive comments, which have contributed significantly to improving the clarity, rigor, and overall quality of our manuscript.
All the reviewer’s comments have been carefully considered and addressed in the revised version of the manuscript. We have made corresponding modifications throughout the text. These changes included (1) the statistical model was clarified and expanded; (2) the structural limitations of the forage, the limited legume contribution, and its implications for milk production and methane emissions were highlighted; (3) a paragraph to support our findings within similar pasture-based systems was included (Line 237 to 248 in the revised manuscript).
We believe the revisions have substantially improved the manuscript, and we are grateful for your valuable input.
Round 2
Reviewer 4 Report
Comments and Suggestions for Authors
The revised version of the manuscript properly addresses yhe comments and suggestions made in the previous review. The imporvements in clarity and scientific soundness have enhanced the overall quility of the work. I consider the current version suitable for publication.
Author Response
Comments 1: The revised version of the manuscript properly addresses yhe comments and suggestions made in the previous review. The imporvements in clarity and scientific soundness have enhanced the overall quility of the work. I consider the current version suitable for publication.
Response 1: Thank you very much for your positive evaluation of our revised manuscript. We sincerely appreciate your constructive feedback, which helped us improve the clarity and scientific rigor of the work. We are pleased to know that you now consider the manuscript suitable for publication.